# Maturation of the Visceral (Gut-Adipose-Liver) Network in Response to the Weaning Reaction versus Adult Age and Impact of Maternal High-Fat Diet

**DOI:** 10.3390/nu13103438

**Published:** 2021-09-28

**Authors:** Maria Angela Guzzardi, Federica La Rosa, Daniela Campani, Andrea Cacciato Insilla, Vincenzo De Sena, Daniele Panetta, Maurizia Rossana Brunetto, Ferruccio Bonino, Maria Carmen Collado, Patricia Iozzo

**Affiliations:** 1Institute of Clinical Physiology, National Research Council (CNR), 56124 Pisa, Italy; m.guzzardi@ifc.cnr.it (M.A.G.); larosa.fed@gmail.com (F.L.R.); vincenzo.desena@isa.cnr.it (V.D.S.); daniele.panetta@ifc.cnr.it (D.P.); 2Department of Surgical, Medical, Molecular Pathology and Critical Care Medicine, Division of Pathology, Pisa University Hospital, 56124 Pisa, Italy; daniela.campani@med.unipi.it (D.C.); andrea.cacciatoinsilla@gmail.com (A.C.I.); 3Department of Clinical and Experimental Medicine, University of Pisa, 56124 Pisa, Italy; maurizia.brunetto@unipi.it; 4Department of Medical Specialties and Hepatology Unit and Laboratory of Molecular Genetics and Pathology of Hepatitis Viruses, Pisa University Hospital, 56124 Pisa, Italy; 5Institute of Biostructure and Bioimaging (IBB), National Research Council (CNR), 80145 Napoli, Italy; ferruccio.bonino@unipi.it; 6Institute of Agrochemistry and Food Technology-National Research Council (IATA-CSIC), 46980 Valencia, Spain; mcolam@iata.csic.es

**Keywords:** maternal obesity, weaning reaction, gut microbiota, KEGG pathways, positron emission tomography, computerized tomography, glucose metabolism, inflammation, liver steatohepatitis, TNF-alpha

## Abstract

Metabolic-associated fatty liver disease is a major cause of chronic pathologies, of which maternal obesity is a frequent risk factor. Gut wall and microbiota, visceral fat, and liver form a pre-systemic network for substrates and pro-inflammatory factors entering the body, undergoing accelerated maturation in early-life when the weaning reaction, i.e., a transitory inflammatory condition, affects lifelong health. We aimed to characterize organ metabolism in the above network, in relation to weaning reaction and maternal obesity. Weaning or 6-months-old offspring of high-fat-diet and normal-diet fed dams underwent in vivo imaging of pre-/post-systemic glucose uptake and tissue radiodensity in the liver, visceral fat, and intestine, a liver histology, and microbiota and metabolic pathway analyses. Weaning mice showed the dominance of gut *Clostridia* and *Bacteroidia* members, overexpressing pathways of tissue replication and inflammation; adulthood increased proneness to steatohepatitis, and *Desulfovibrio* and *RF39* bacteria, and lipopolysaccharide, bile acid, glycosaminoglycan, and sphingolipid metabolic pathways. In vivo imaging could track organ maturation, liver inflammation, and protective responses. A maternal high-fat diet amplified the weaning reaction, elevating liver glucose uptake, triglyceride levels, and steatohepatitis susceptibility along the lifespan. The visceral network establishes a balance between metabolism and inflammation, with clear imaging biomarkers, and crucial modulation in the weaning time window.

## 1. Introduction

Due to a pandemic spread of obesity, metabolic-associated fatty liver disease and hepatic inflammation have become the most frequent causes of liver-related and metabolic co-morbidity in adults and children [1]. Primary prevention is the only way to circumvent the current lack of effective therapy and of screening tools able to capture individuals at risk of progression. Among early, modifiable, and most common risk factors, maternal obesity has a life-long impact on body weight and gut microbiota composition, programming metabolic disorders in the offspring [2]; however, its direct contribution to liver disease remains controversial. Some studies indicate that a maternal high-fat diet (HFD) is sufficient [3], whereas others indicate that post-natal HFD in the offspring is necessary to determine severe liver damage [4,5]. In addition, one study indicates that a mismatch of normal diet (ND) in mothers and HFD in offspring is the only triggering combination [6].

The liver is located in a strategic position to handle substrates, metabolites, and pro-inflammatory factors deriving from the gut and visceral fat, and regulates their access to the systemic circulation. In fact, these three organs receive blood directly from mesenteric or portal routes, thereby establishing an internal (pre-systemic) anatomical and functional network before connecting with the rest of the body. They share a primary role in triggering/blunting inflammatory states [7,8,9,10,11], and they determine the fate of endogenous and exogenous metabolic substrates. Their composition and function undergo important changes during early growth, including the hematopoietic-to-metabolic transition of the liver [12], the thickening and replication of intestinal glands, with immune cell and microbiota colonization [13,14], and the proliferation and differentiation of (pro)adipocytes via hyperplasia [15], which typically occurs in early life. In particular, the weaning period represents a critical time window for the programming of a life-long cooperation between the gut, visceral fat, and liver, as the dramatic and rapid bacterial changes caused by weaning provoke a benign inflammatory reaction, imprinting the immune system, and reducing the likelihood of chronic and inflammatory diseases in adult life [14,16,17,18]. During the life course, these organs continue to co-modulate the risk of systemic and liver dysmetabolism. Each of them has been implicated in the pathogenesis of chronic metabolic diseases, but their cross-talk along maturation windows remains to be elucidated, together with the impact of maternal obesity. Understanding these relationships will translate into diversified, mechanism-targeted prevention strategies and time windows. Multimodal imaging of abdominal organs by, e.g., positron emission tomography (PET) and computerized tomography (CT) attenuation measures (reflecting organ structure, fibrosis, lipid content) offers the unique possibility to address relationships between gut, liver, and visceral fat in vivo. This has already revealed that tissue glucose uptake plays a role in triglyceride accumulation and release (16), that intestinal glucose uptake is impaired in metabolic diseases [17], and that visceral fat and liver glucose metabolism are correlated in adult humans [18], in whom adipose tissue may serve as a sink against liver substrate overload [19], at the expense of adipocyte enlargement and low-grade systemic inflammation.

In the current study, we hypothesized that the simultaneous evaluation of glucose handling and tissue radiodensity in interactive abdominal organs in the context of the weaning reaction and in adulthood, as well as in high- vs. low-risk mice, would disclose new insights in the pathogenesis and indicators of metabolic and liver damage, its potential modulators or primary prevention targets, and time windows. We conjectured that the pattern of microbiota maturation and its location (caecum versus colon) could participate in the regulation of the visceral cross-talk. To explore these hypotheses, in vivo imaging of pre-systemic and post-systemic glucose extraction and glucose uptake (by PET and ^18^F-labelled fluorodeoxyglucose: ^18^FDG) and tissue radiodensity (by CT) in the liver, visceral fat, and intestine in combination with liver histology and microbiota and a metabolic pathway KEGG analysis (caecum and colon) were carried out in the offspring of HFD and ND dams at the ages of weaning or 6 months.

## 2. Materials and Methods

### 2.1. Study Design

B6129SF2/J (stock no: 101045, The Jackson Laboratory, Bar Harbor, Maine) female mice underwent ND (11% kcals from fat, Mucedola s.r.l., Milan, Italy, *n* = 5), or HFD (58% kcals from fat, Mucedola, *n* = 4) for 3 months before mating, and during gestation and lactation. After weaning, offspring (total *n* = 38) were fed with a standard diet. Animals were housed under standard conditions (22 °C, 12-hour light/dark cycles), with ad libitum access to food and water. Food intake, body weight, and glycemia have been already reported together with cognitive data [2]. At weaning (*n* = 19, NDoff *n* = 11, HFDoff *n* = 8) or at 6 months of age (adulthood, *n* = 19, NDoff *n* = 10, HFDoff *n* = 9), liver, visceral fat, and intestinal glucose uptake and density were measured by PET-CT imaging. Then, animals were euthanized by anesthetic overdose, and liver biopsies, blood, and fecal samples (colon, caecum) were collected.

### 2.2. Tissue Metabolism

Imaging of ^18^FDG was performed under fasting conditions (PET-CT IRIS, Inviscan SAS, Strasbourg, France), under isofluorane anesthesia (IsoFlo^®^, Abbott Laboratories, Chicago, IL, USA). Breath frequency and temperature were monitored during the study and a heated pad was used to prevent hypothermia. ^18^FDG was administered by intraperitoneal (i.p.) injection to enable first-pass glucose uptake via mesenteric and portal vein vessels in visceral organs (pre-systemic), before entry into/delivery from the general circulation (post-systemic glucose uptake). A 60-minute whole-body dynamic PET scan was performed, and glycemia was measured in tail blood by a glucometer (OneTouch, Johnson&Johnson Medical SpA, Pomezia, Italy). PET data were corrected for dead time, random coincidences, and radioactive decay, and reconstructed by a 3D-Ordered Subset Expectation Maximization (3D-OSEM) algorithm. CT images where corrected for beam hardening and ring artifacts, reconstructed with cone-beam filtered backprojection (FBP), and calibrated in Hounsfield units (HU). All PET and CT images were exported to DICOM format after reconstruction, and fused within the AMIDE Medical Image Data Examiner 1.0.4 (http://amide.sourceforge.net/, accessed on 2020). Regions of interest were manually drawn on PET-CT images in correspondence with the right liver lobe, the large intestine, and lower abdominal visceral fat. Tissue radiodensity in HU (from now on defined as density) was extracted from the CT images in corresponding areas. Tissue time activity curves were normalized to the injected dose per gram of body weight (%ID/g), representing the glucose fractional extraction ( Appendix AA–C), and multiplied by glycaemia to estimate glucose uptake [2]. Glucose extraction and uptake rates were integrated over the first 10 min to primarily reflect organ pre-systemic glucose entry, and in the subsequent 10–60 min interval to reflect post-systemic glucose extraction. This was supported by the observation ( Appendix A) that in the first 10 min, left ventricular blood had received 4% of the total amount of tracer entering the general circulation in the entire imaging period, with 96% of ^18^FDG reaching the general circulation in the following 10–60 min. Instead, 15–20% of the tracer reached the gut and visceral fat very rapidly, followed by the liver, within the pre-systemic 10 min time window. The systemic clearance of glucose was computed as a ratio of injected to integrated blood ^18^FDG activity, and multiplied by glucose levels to reflect the endogenous glucose production (EGP) [20], as expressed per gram of body weight.

### 2.3. Biochemical Analyses

Blood samples were collected at the end of the imaging procedures for a plasma or serum biochemical analysis. Triglycerides and liver enzymes (aspartate aminotransferase, AST, alanine aminotransferase, ALT) were determined by a bench clinical chemistry analyzer (Reflovet^®^ Plus, scil animal care company S.r.l., Treviglio, Italy) in 28–30 high quality samples. Triglyceride results falling below the measurable range were equaled to the lowest value of the accessible range (70 mg/dl). Inflammatory markers were measured by Luminex^®^ xMAP^®^ technology (Merck-Millipore Corp., Boston, MA, USA); they have been previously reported [2], and were only used here to confirm the occurrence of the weaning reaction and examine associations with the metabolism of visceral organs.

### 2.4. Liver Histology

Tissue samples for histology were collected and processed in half of the cases (*n* = 19). They were fixed in 10% formalin for 24 h, dehydrated, and included in paraffin using the Donatello Diapath automatic tissue processor (Martinengo, Bergamo, Italy), sliced (HistoCore Autocut, Leica BioSystems microtome) with 2 μm thickness, and stained with hematoxylin and eosin using the automated Dako CoverStainer (Santa Clara, CA, USA). Each section was documented at 20× and 40× magnification, by using the Olympus BX51 microscope and connected with an Olympus DP70 digital camera and AnalySIS 5.0 imaging system software (Olympus, Tokyo, Japan). Analyses were adapted from the method of Kleiner et al. [21] to evaluate and score (yes/no, 1/0) vessel dilatation, fibrosis, portal and lobular inflammation (also graded by foci number at 20× magnification, 1 = one focus, 2 = two-four foci, 3 = >four foci), or ballooning degeneration, and micro/macrovesicular steatosis (also graded as the percentage of the affected cells). The diagnosis of non-alcoholic steatohepatitis was defined as the sum of grades ≥4 (sum score: range 4–7, including steatosis, lobular inflammation, and ballooning), and categorically (non-alcoholic steatohepatitis, NASH yes/no).

### 2.5. Gut Bacteria 16S rRNA Gene Sequencing

Total DNA was isolated from the caecum and colon fecal pellets by the MasterPure Complete DNA&RNA Purification Kit (Epicentre, Illumina, San Diego, WI, USA), as previously described [22]. DNA was measured using a Qubit^®^ 2.0 Fluorometer (Life Technology, Carlsbad, CA, USA) and normalized to 10 ng/μL. The V3-V4 region of the 16S rRNA gene was amplified by PCR using Illumina adapter overhang nucleotide sequences according to Illumina protocols. The multiplexing step was performed using a Nextera XT Index Kit (Illumina, San Diego, CA, USA). A Bioanalyzer DNA 1000 chip (Agilent Technologies, Santa Clara, CA, USA) was used to check the PCR product, and libraries were sequenced using a 2 × 300 bp paired-end run (MiSeq Reagent kit v3) on a MiSeq-Illumina platform (FISABIO sequencing service, Valencia, Spain) according to the manufacturer’s instructions (Illumina). For quality control, reagents employed for DNA extraction and PCR amplification were also sequenced. A quality assessment was performed by the prinseq-lite program (min_length: 50; trim_qual_right: 20; trim_qual_type: mean; trim_qual_window: 20 [23]). R1 and R2 from sequencing were joined using fastq-join from ea-tools suite (http://code.google.com/p/ea-utils, sequences from 2016–2017). Data were obtained using an ad-hoc pipeline written in R Statistics environment and data processing was performed by a QIIME pipeline (version 1.9.0) [2,24]. Chimeric sequences and sequences that could not be aligned were removed. The clustered sequences were utilized to construct OTUs tables (97% identity), then classified into phylum, family, and genus taxonomic levels based on the Greengenes database v13.8. Sequences not taxonomically classified or belonging to cyanobacteria and chloroplasts (representing ingested plant material) were removed.

### 2.6. Statistical Analysis

The results were first analyzed by age groups, and then stratified by maternal habitus*age, under the rationale given in the results section. The metabolic profile, imaging results, and circulating markers were compared between groups with the analysis of variance or the Student’s t-test, as appropriate. Standard regression analyses were used to evaluate associations. These results are presented as mean ± sem, and *p* values ≤0.05 were regarded as statistically significant. Relative microbial abundances were obtained with the Calypso pipeline tool, version 8.84, by using total sum normalization. A redundancy analysis (RDA) was used to assess the complex associations of gut bacteria community composition in the offspring. A linear discriminant analysis (LDA) and effect size (LEfSe) analyses were used to identify unique biomarkers (LDA score > 3.0) in relative abundance of bacterial taxonomy [25]. Spearman’s correlation analysis was used to explore univariate associations between bacteria taxa relative abundances and imaging parameters. A predictive inferred functional analysis was performed using PICRUSt with the Kyoto Encyclopedia of Genes and Genomes (KEGG) [26]. Then, the LEfSe analysis and Wilcoxon-rank test were used to explore associations between KEGG pathways and groups. A false discovery rate (FDR) correction was applied in multiple comparisons/associations analyses.

## 3. Results

Results are first presented by age periods (pooling HFDoff and NDoff at each age), to reflect the maturation of the gut–visceral fat–liver axis in a general population, with the expected prevalence of being overweight and the frequency of individuals born to overweight mothers seen in a general population. Results are then stratified by maternal group at each age to dissect the impact of early-life dietary exposures on the development of the visceral network and metabolic disorders.

### 3.1. Effect of Age on Systemic and Tissue Metabolism

Maternal body weight was similar between the pooled age groups. Adult offspring were expectedly heavier than weaning mice (31 ± 2 vs. 17 ± 1 g, *p* < 0.0001), whereas circulating glucose and triglyceride (TG) levels did not differ significantly (n.s.). Weaning was characterized by a pronounced proinflammatory reaction at the systemic level, as suggested by high circulating concentrations of TNFα, IL6, and PAI-1, which were markedly reduced in adults (Figure 1A). Adults were characterized by a greater CT density in the liver and gut wall (Figure 1B), higher frequency of hepatic lobular inflammation, higher steatohepatitis scores in biopsies (Figure 1C), and greater systemic glucose clearance and EGP (Figure 1D). Comparing the kinetics of the glucose tracer among the three tissues (Appendix A), visceral fat and gut were more glucose avid at weaning, whereas the liver became more glucose avid in adult life. This was due to a marked reduction in pre-systemic and post-systemic glucose extraction and uptake in the intestine and visceral fat, as opposed to a mild decline only involving pre-systemic glucose extraction and uptake in the liver (Figure 1E, Appendix A). As a result, the visceral fat/liver and gut/liver glucose uptake ratios (0–60 imaging time-range) were three times higher in weaning than adult mice, and conversely the relative amount of glucose going into the liver (liver-to-visceral fat and liver-to-gut partitioning) was more than doubled in adults than weaning mice (Figure 1F).

### 3.2. Effect of Maternal Diet on Systemic and Tissue Metabolism

In comparing the weaning reaction in mice born to HFD vs. ND mothers (Figure 1H–L, Appendix A), the increase in TNFα, PAI-1, and intestinal glucose extraction and uptake was present in both groups (compared to adult age), but HFDoff tended towards higher levels of IL6, together with a two-to-six-fold elevation in glucose extraction and uptake in visceral fat and (less markedly) in the gut than NDoff. These differences were not seen in adults. Instead, adult HFDoff, were characterized by a 40% elevation in post-systemic hepatic glucose uptake compared to NDoff (Figure 1L). Consequently, in HFDoff the visceral fat/liver (glucose uptake) ratio was very high at weaning (+300%) compared to NDoff (Figure 1M). These patterns suggest that visceral fat and the gut received and sequestered more glucose in the weaning reaction of HFDoff than NDoff, reducing the within-network (i.e., pre-systemic) flux of glucose to the liver, preserving fasting EGP (n.s., not shown) and glucose clearance (Figure 1K). This pre-systemic glucose-sinking effect was lost in adult HFDoff, resulting in a high post-systemic liver glucose uptake and a lack in the physiological increase in systemic glucose clearance observed in NDoff. In addition, the CT density (Figure 1I) in the visceral fat and liver was high in weaning HFDoff, thereby failing to undergo the physiological increase from weaning-to-adulthood, as seen in NDoff. Biopsy data in the available samples demonstrated a significant trend of steatohepatitis grades, worsening across the four age*maternal-diet subgroups (Figure 1J), supporting a summative contribution of age and maternal HFD, in which adult HFDoff had the most severe liver condition with a 40% elevation in circulating triglyceride levels (1.15 ± 0.10 vs. 0.79 ± 0.00 mmol/L, *p* < 0.003 vs. adult NDoff). This group also lacked the physiological elevation in MCP1 levels observed in age-matched NDoff, thereby showing a four-fold cytokine deficiency (Figure 1H), as previously reported [2].

### 3.3. Tissue Metabolism Associates with Systemic Inflammation, Liver Steatosis, and Steatohepatitis

Correlations between imaging markers and inflammatory cytokines or histological liver results were analyzed over the full range of pooled groups to interpret the CT and PET findings (Figure 1G). Results of these analyses documented that the visceral fat metabolism was the strongest indicator of systemic inflammation. Pre-systemic liver glucose extraction and uptake, and pre-/post-systemic intestinal and visceral fat glucose extraction and uptake were positively associated with TNFα and IL6, and (the former two) with PAI-1. Higher pre-systemic hepatic glucose extraction (*r* = −0.35, *p* = 0.03) and pre-/post-systemic gut glucose extraction (*r* = −0.33, *p* = 0.05, *r* = −0.38, *p* = 0.02) were also predictors of the lowering of hepatic CT density, which was a valid in vivo indicator of histology proven macrovescicular steatosis (*r* = −0.50, *p* = 0.03). The visceral fat CT density was also strongly and negatively related to liver macrosteatosis (*r* = −0.61, *p* = 0.009), whereas increasing liver CT density was proportional to biopsy-proven lobular inflammation (*r* = 0.42, *p* = 0.07). Higher pre-systemic liver glucose extraction and uptake were positive predictors of portal inflammation (*r* = 0.48, *p* = 0.036, *r* = 0.46, *p* = 0.05), whereas higher post-systemic visceral fat glucose uptake (*r* = −0.57, *p* = 0.017) and colon glucose uptake (*r* = −0.52, *p* = 0.03) were negative predictors of lobular inflammation. Portal inflammation (*r* = −0.49, *p* = 0.03) was negatively related to liver steatosis. The intestinal wall CT density was inversely related to post-systemic gut glucose extraction and uptake (*r* = −0.37, *p* = 0.025, *r* = −0.36, *p* = 0.031). The visceral fat CT density was positively related to post-systemic visceral fat glucose extraction (*r* = 0.35, *p* = 0.039). The above findings suggest that liver glucose overexposure, especially via the portal route (pre-systemic) may subtend a risk of inflammation and steatohepatitis, which can be partly counterbalanced by the ability of visceral fat and gut to capture glucose.

### 3.4. Microbiota and Metabolic Pathway Analyses

Among the three visceral tissues, microbiota composition and metabolic pathways were more frequently and more strongly related to imaging parameters describing the gut, followed by the liver, and then visceral fat. Caecum microbiota and metabolic pathways seemed more influential than their respective colon counterparts. In the current KEGG evaluation, we focused on functions related to substrate metabolism and inflammation, excluding pathways more strictly related to bacterial constituents (e.g., RNA, ribosome, DNA), cofactors-vitamins, or drugs and toxicants.

#### 3.4.1. Effect of Age

The RDA analyses showed highly significant differences in the microbiota profile between weaning and adult age in both the caecum and colon (Figure 2A). Explaining this difference, LDA analyses and LEfSE scores identified a panel of unique biomarkers (Figure 2B). The weaning microbiota was typified by high abundance in taxa belonging to the *Bacteroidia* (Bacteroidetes phylum) and the *Clostridia* classes (Firmicutes phylum), and the *Biophila* genus (Proteobacteria phylum) in the caecum, whereas only the *S24-7* family/genus (also *Muribaculaceae/Muribaculum*) were enriched in the weaning colon. In adults, taxonomic biomarkers belonged to the Actinobacteria (only caecum), Proteobacteria, Tenericutes, Verrucomicrobia phyla (both caecum and colon), *Porphyromonadaceae*, and *Rikinellaceae* families (Bacteroidetes phylum, only colon), several taxa belonging to *Bacilli* order (including *Bacillus*, *Streptococcus*, *Unclassified Planococcaceae* in both caecum and colon, or *Lactobacillus* genera in caecum), and *Blautia* and *Coprobacillus* genera (only caecum) (Firmicutes phylum). Considering the significant increase in NASH susceptibility observed in the adult age, we conducted targeted multivariant RDA and LDA-LEfSE analyses (Figure 2D–E) to identify unique biomarkers of NASH, and found that NASH-positive mice were discriminated by high abundance in *Desulfovibrio* genus in the colon, and *Aerococcus* genus and *Aerococcaceae* family (belonging to *Bacilli* class, Firmicutes phylum) and *Unclassified RF39* genera, Tenericutes phylum, Mollicutes class, *RF39* order, *Unclassified RF39* families in the caecum.

Most tissue parameters measured in vivo and ex vivo in the liver, gut, and visceral fat showed opposite correlations with weaning- vs. adulthood-dominating bacteria, suggesting that the changing microbiota composition is strictly aligned to, and may explain the changes occurring in the metabolism and structure of these tissues during maturation. In Figure 3A,B and Appendix A, we restricted the selection only to the tissue parameters that were significantly different between ages, and the bacteria (genus level) and metabolic pathways showing significant correlations with these parameters. Among them, the increase in *Adelcreutzia*, *Lactobacillus*, *RF39*, *Desulfovibrio*, *Bacillus*, and *Akkermansia* genera, and the decrease in *S24-7*, *Oscillospira*, *Ruminococcus*, and *RC44* genera, especially in the caecum, from weaning to adulthood were associated with the corresponding increase in tissue density and decrease in glucose uptake observed between ages. In addition, the increase in *Desulfovibrio*, *RF39*, and *Coprobacillus* genus abundance were predictive of greater liver inflammation, which is consistent with the results of the LDA/LEfSE analysis in Figure 2E. Metabolic pathways involved in proinflammatory imprinting (lipopolysaccharide-LPS, sphyngolipids, MAPK signaling) and in organ proliferation and maturation (energy and lipid metabolism, glycan production, bile acids, and several amino-acids biosyntheses) were high at weaning (Figure 2C and Figure 3B), whereas glycolysis-gluconeogenesis, the phosphotransferase system, amino-acid (tyrosine, alanine), and sulfur related pathways were hallmarks of adult age. Correlations between the expression level of these pathways and organ density or metabolism were coherent with the changes occurring in these imaging biomarkers from weaning to adulthood. The bile acids, LPS, sphingolipid, glycosaminoglycan degradation, and amino-acid metabolic pathways were positively correlated with liver inflammation or were unique identifiers of NASH, whereas the caecum glutamine-glutamate pathway seemed protective (Figure 2F and Figure 3B).

#### 3.4.2. Effect of Maternal Diet

Microbiota richness decreased significantly in the caecum from 1 to 6 months of age only in adult HFDoff. We have previously reported a description of differently abundant taxa in HFDoff and NDoff, together with KEGG pathways in the colon [2]. In the current study, we extended KEGG analyses to the caecum, and we presented differences in the pathways related to substrate metabolism and inflammation in both the caecum and colon ( Appendix A). In Figure 3C–F and Appendix A, we selected the tissue parameters that were significantly different between age-matched HFDoff and NDoff, and the bacteria (family and genus level) and metabolic pathways showing significant correlations with these parameters. These involved an enrichment in *Clostridia* and *Bacteroidia* classes, and the Tenericutes-*RF39* genus, and a depletion in the *Anaerotruncus* genus in weaning HFDoff vs. NDoff, correlating with the observed metabolic features characterizing HFDoff at this age (increased visceral fat and intestinal glucose uptake). In adult HFDoff, showing high liver and intestinal glucose uptake (vs. NDoff), the associated biomarkers were *Desulfovibrio* and *Allobaculum* genera, *Clostridiales* order (*Unclassified Clostridiales* genera, *Peptococcaceae* family), and *Coriobacteriaceae* family. In the KEGG analysis, only three amino-acid related pathways (lysine, phenylalanine, tryptophan) were simultaneously associated with the maternal diet and with the tissue differences observed at weaning. In adults, the upregulation of insulin signaling and downregulation of nitrogen metabolism and butyrate (short-chain fatty acid) pathways detected in HFDoff were predictors of histological liver inflammation, whereas pathways regulating arachidonic acid, glycan, sulfur, and amino-acid functions were associated to the imaging outcomes.

## 4. Discussion

The current study documents that maturation from weaning to adulthood leads to profound changes in the metabolism and structure of visceral tissues and in microbiota composition and function, with an increased susceptibility to liver inflammation. Our results identify clear in vivo imaging indicators of organ maturation, namely tissue glucose uptake and organ density, showing strict associations with changes in microbiota composition and metabolic pathway expression, modulating plasma levels of inflammatory markers. We provide evidence that the weaning reaction is amplified, and organ metabolism is affected in mice born to HFD compared to ND mothers, in line with microbiota characteristics, which may contribute to the metabolic abnormalities observed especially in adult age, i.e., being overweight, hyperglycemia, hypertriglyceridemia, and liver inflammation. The suggested sequence of events is exemplified in Figure 4.

Our data were first examined across age periods to reflect the maturation of the gut–visceral fat–liver axis in a general population, and then addressed in relation to maternal diet. As a general finding, caecum compared to colon microbiota showed stronger connections with our imaging and histology measures, underlining its role in substrate exchange and immunity development.

### 4.1. Effects of Age: Maturation of the Visceral Network

Maturation of the intestinal wall is concentrated in early life and implies crypt fission, goblet cell proliferation, mucus thickening, and impermeability, while adipose tissue undergoes differentiation and hyperplasia, and the liver undergoes the transition from being a predominantly hematopoietic to the most important metabolic organ, with a growing ability to produce glucose, replacing maternal sources [12,13,14,15,27,28,29]. In coherence with these changes, we found a significant increase in hepatic density from weaning to adult age, and the liver became the major (in adulthood), from being the minor (at weaning) glucose consumer per unit of tissue volume compared to gut and visceral fat, also increasing its release of glucose (EGP) in proportion to body weight and systemic glucose clearance. Notably, bacteria (especially *S24-7*) and metabolites (i.e., LPS, bile acids, amino-acids, energy metabolism) that were dominant at weaning in our mice have been implicated in liver regeneration processes after liver resection, co-regulating the proliferative and metabolic roles of the liver [30]. Similar to the liver, the immature gut was characterized by low wall density and high glucose requirements, signaling an active proliferation. Microbiota-induced metabolic pathways (energy metabolism, oxidative phosphorylation, glycan, glycosamine, sphingolipid, glycosphingolipid metabolism, adipocytokine signalling, peroxisomes, branched-chain, and other amino acids) related to energy-fueling, protein-building, and mucin-promoting functions prevailed in this period, and were strongly predictive of the intestinal imaging hallmarks, establishing a coherent mechanistic scenario. Finally, visceral fat showed a high glucose uptake at weaning, consistent with the hyperplastic phase, and a (negative) CT density (Hounsfield unit) range indicating a prevailing lipid occupancy. The visceral fat glucose uptake was associated with the abundance in *Oscillospira* (*Clostridiales* order) and *RC44* genera, and with metabolic pathways involving peroxisome, oxidative phosphorylation, and energy metabolism, beyond ammino-acid biosynthesis.

One important phenomenon occurring during our early-life observation period is the weaning reaction, i.e., a benign proinflammatory reaction, consisting of microbiota induced elevations in TNFα levels in the circulation, and an immature impermeability of the developing intestinal barrier, playing a permissive role on microbiota interactions with visceral organs [31,32,33]. The reaction lasts two weeks in mice and impacts a number of disease conditions during life. Consistent with this, we observed a several-fold elevation in TNFα, IL6, and PAI-1 levels in our weaning mice, compared to adults. Their microbiota was typified by a high abundance in LPS producing *Bacteroidia* (*Rikenellaceae* in caecum, *S24-7* in caecum and colon) and *Clostridiales* genera, and an upregulation in LPS biosynthetic and MAPK signaling pathways in the KEGG analysis, all of which have been associated with the weaning reaction, including hepatic oxidative stress [31,34]. Older studies have shown that the infusion of LPS results in a pronounced increment in tissue glucose uptake, due to the build-up of glucose-avid immune cells, with the greatest response occurring in the liver [35]. On the same principle, ^18^FDG-PET imaging has been used to capture inflammation in adipose tissue [36], and permeability in the gut wall [37]. Notably, visceral fat plays an important role in generating cytokines, and visceral fat glucose uptake showed the strongest correlation with the levels of circulating TNFα and IL6 in this study, followed by gut and liver glucose uptake and CT densities, whereas PAI-1 was preferentially related to hepatic pre-systemic glucose uptake. In fact, pre-systemic liver glucose uptake was high in our weaning mice, correlating with the degree of biopsy-proven portal inflammation, which is typically seen in pediatric liver dysmetabolism [38]. Interestingly, pre-systemic liver glucose uptake was also the only metabolic process associating with the abundance of all bacteria that were dominant at weaning, and showing a significant association with the LPS biosynthetic pathway.

In synthesis, weaning mice displayed a very high tissue glucose uptake resulting from rapid proliferation and a marked proinflammatory reaction; the structural immaturity of the gut and liver tissues translated in low CT density. Both imaging hallmarks are closely associated with the microbiota and its metabolic pathways. The pre-systemic liver glucose uptake was reflective of biopsy-proven portal inflammation and associated to the microbiota induced LPS pathway. The three folds higher visceral fat/liver and gut/liver glucose uptake ratios seen in this life-phase are consistent with the role of the gut and visceral fat in generating and buffering the proinflammatory reaction, and may protect the liver from more severe substrate overload and damage, as suggested by the inverse correlations observed between visceral fat or gut glucose uptake and biopsy-measured liver inflammation.

Maturation towards adulthood involved an increase in tissue density and a major decrease in pre-/post-systemic glucose uptake especially in visceral fat and the gut, with higher EGP and whole-body glucose clearance. Bacterial composition was significantly shifted away from the dominance of *Bacteroidia* and *Clostridia* classes during weaning, in favor of other phyla or classes, namely Actinobacteria, Proteobacteria (*Desulfovibrio* genus), Tenericutes (*RF39* genus), Verrucomicrobia (*Akkermansia*), Bacteroidetes (*Porphyromonadaceae*, *Rikinellaceae* genera), and Firmicutes (*Bacilli*, *Blautia* genera). In line with the transient nature of the weaning reaction, systemic inflammatory markers were expectedly declined, and the decrease was correlated with the changes in organ metabolism and structure. The relative abundance of Verrucomicrobia phylum characterized by the presence of *Akkermansia* genus was associated with the lowering of pre-systemic tissue glucose uptake, most strongly in visceral fat, which was dependent on the reduction of systemic inflammation. The *Akkermansia* genus has been proven to reduce systemic inflammation, by suppressing LPS, TNFα, and other cytokine levels and increasing fatty acid oxidation in adipose tissue, as reviewed in [39]. This previous knowledge provides mechanistic support to the observations of the current study, also suggesting that *Akkermansia* should be investigated as a potential factor terminating the weaning reaction. As opposed to a decline in systemic inflammatory markers, we observed an increased susceptibility of the liver to develop steatohepatitis in adult mice. The inverse association between post-systemic glucose uptake in visceral fat and gut (reduced in adults) and lobular liver inflammation (increased in adults) suggests that a diminished sequestration of glucose in these extra-hepatic tissues may detract protection against oxidative stress and liver inflammation. In fact, the sum of glucose release and uptake by the liver indicates that the inflow of glucose and gluconeogenic substrates into the liver was increased in adult mice. It is of note that the reduction in visceral fat and gut glucose uptake in adults was proportional to the abundance of *Desulfovibrio* and *RF39* genera (i.e., the two main predictors of NASH) and to the upregulation in the sulfur relay system and glycolysis-gluconeogenesis pathways in the KEGG analysis. In fact, the emerging roles of hydrogen sulfide point towards a negative effect on glucose uptake in adipocytes [40] and a stimulatory effect on hepatic pyruvate carboxylase sulfhydration, promoting gluconeogenesis in liver cells [41]. Hence, the upregulation of the sulfur relay system and glycolysis-gluconeogenesis pathways might underlie the reduction in visceral fat and gut glucose uptake, as well as the increase in EGP observed in our adult mice.

In synthesis, the above findings suggest that the increase in *Akkermansia* genus abundance may be involved in the downregulation of systemic inflammation and the associated pre-systemic glucose uptake, especially in visceral fat, whereas the upregulation of *Desulfovibrio* and *RF39* bacteria may activate metabolic pathways linked with liver inflammation susceptibility. Among these, our data point towards the (in)ability of visceral fat and gut to extract glucose from the systemic circulation (post-systemic glucose uptake), leading to liver glucose overexposure and an increase in EGP.

The immune modulatory impact of the gut microbiota can significantly affect intrahepatic and intestinal inflammation, and hepatocarcinogenesis. Our findings are consistent with the evidence that patients with cirrhosis and steatohepatitis who developed hepatocellular carcinoma (HCC) lacked similar protective bacteria profiles and had enhanced liver and intestinal inflammation as compared to patients with cirrhosis, but without HCC [42]. Cirrhosis patients had a lower abundance of *Akkermansia* genus abundance than controls in association with an inflammatory intestinal environment and higher fecal calprotectin levels, and *Bifidobacterium* was also depleted in HCC patients [43]. Thus, the combined deficiency of these beneficial bacteria might enhance intestinal and liver inflammation, influencing liver disease progression as well as the initiation and/or progression of hepatocarcinogenesis, and the administration of probiotics was shown to reduce the incidence and growth of HCC lesions in mice, by modulating gut microbiota composition and restoring intestinal permeability [44,45,46].

### 4.2. Effects of Maternal Diet*Age on the Visceral Network

We asked to which extent does an early exposure to ND or HFD contribute to, or deviate from, the results obtained in a more general, i.e., admixed population. The microbiota of weaning HFDoff showed an amplified abundance of *Bacterioidia* and *Clostridia* class bacteria, namely *Rikenellaceae*, *Parabacteriodes*, and *Peptostreptococcaceae* genera, and in *RF39*. In line with this, visceral fat glucose uptake was several folds higher in HFDoff than NDoff at weaning; the gut was affected to a lower extent, and a trend towards higher IL6 levels in HFDoff was seen. The liver showed a greater CT density in this group. Interestingly, both *RF39* abundance and lobular inflammation were positive predictors of hepatic CT density, and an LDA analysis indicated that *RF39* was a significant discriminator of biopsy-proven NASH. We screened for functional pathways that could mediate the interactions between microbiota and tissue outcomes, and we identified three (upregulated) metabolic pathways that were associated (positively) with liver CT density and/or visceral fat glucose uptake, namely tryptophan metabolism, lysine degradation, and phenylalanine metabolism. Lysine degradation was the most upregulated (+15–30%) pathway in both the colon and caecum. Lysine can be processed by *Clostridia* to generate short-chain fatty acids (SCFAs) [47], and we correspondingly observed an upregulation of fatty acid and butanoate (butyric acid) metabolic pathways in the colon and caecum. SCFAs also affect tryptophan metabolism, which has major effects on intestinal barrier integrity and transit, and fatty liver disease and metabolism [48]. In states of obesity and inflammatory bowel diseases, the metabolism of tryptophan is enhanced in the adipose tissue, correlating with IL6 levels [49] and leading to increased intestinal permeability and LPS translocation in the systemic circulation, resulting in peripheral inflammation, altered glucose metabolism, and impaired hepatic protection [50]. These aminoacidic actions recapitulate very well the findings of the current study, namely the potentiation of cytokine IL6 levels and of visceral fat and gut glucose uptake observed in weaning HFDoff, as well as the tendency towards liver inflammation and the occurrence of hyperglycemia in the fed state, as previously shown in this mice group [2].

We have previously reported that maternal HFD led to adulthood obesity, hyperglycemia, hypoinsulinemia, hyperresistinemia, and impaired MCP1 levels in these mice [2]. In this study, we demonstrated that maternal HFD contributes to increasing the susceptibility towards liver inflammation in the offspring from early to adult life, and provokes a 40% elevation in plasma triglyceride levels in adult HFDoff, limiting the accumulation of tissue lipids. The main difference between adult HFDoff and NDoff at the tissue metabolic level was the upregulation in post-systemic glucose uptake in the liver and (less) in the gut in HFDoff. It is of note that the glucose uptake in these two tissues was related to fasting hyperglycemia, suggesting that visceral fat was unable to act as the substrate sink. As previously shown [2], the microbiota was significantly different in relation to the maternal diet. Among bacteria that were different in adult NDoff and HFDoff, the abundance of *Coriobacteriaceae*, *Desulfovibrio*, and *Allobaculum* were positively associated with liver glucose uptake, and the deficiency of *Clostridiales* and *Peptococcaceae* families in HFDoff predicted higher liver and gut glucose uptake. The abundance of *Desulfovibrio*, *Allobaculum*, and Actinobacteria have been previously shown to reflect the progression of liver disease in Western-diet-fed or HFD-fed diabetic mice [51,52], and the abundance of *Desulfovibrio* genus emerged as a significant LDA-discriminator of biopsy-proven NASH in this study. Consistent with our findings, steatohepatitis in HFD and in overweight (but not in lean) subjects, and in patients with type 2 diabetes was associated with an increase in gut *Desulfovibrio* and sulfate-lowering bacteria [53,54]. We found that arachidonic acid metabolism was augmented by 3.5 folds in adult HFDoff and positively related to gut glucose uptake, together with sulfur and phenylalanine metabolism, and both sulfur reduction and the arachidonic acid cascade have been implicated in gut inflammation, steatohepatitis, and an unhealthy gut–adipose–liver relationship [55,56,57]. We also observed that metabolic pathways of methionine-cysteine (i.e., the only sulfur-containing amino acids) were associated with liver and gut post-systemic glucose uptake. These amino acids have been related to an unhealthy gut–liver–adipose metabolism and to obesity, and their dietary restriction has many benefits, as referenced in [58]. In addition, we noted that the phosphatydilinositol and insulin signaling pathways, and glycan biosynthesis and metabolism were related to liver outcomes in our adult HFDoff mice, and it is of interest that the insulin-phosphatydilinositol kinase pathway has been recently involved in the progression of non-alcoholic liver disease [59,60].

Our results provide a cause–effect demonstration linking weaning, adult age, and maternal diet with respective metabolic, inflammatory, and microbiota outcomes; however, the associations between imaging and microbiota biomarkers support, but do not prove causality. However, a causal relationship was documented in studies addressing microbiota transplantation from 2-week-old infants born to obese (compared to lean) mothers into adult germ-free mice. This led to alterations in intestinal permeability, increased hepatic endoplasmic reticulum stress, and signs of periportal inflammation [38], together with impaired macrophage phagocytic function and cytokine production, the latter promoting weight gain, beyond hepatic macrophage accumulation. These findings are consistent with the overweight, associating with hepatic inflammation susceptibility, and the four-fold MCP1 defect observed in our adult HFDoff.

## 5. Conclusions

Our study is the first to investigate tissue specific CT radiodensity and glucose uptake in vivo, in relation to microbiota composition and function in the context of the weaning reaction, tissue maturation, and maternal diet. The results suggest that pre-systemic hypermetabolism and high CT density reflect inflammatory processes, whereas post-systemic glucose uptake (gut, visceral fat) confers protection against hepatic metabolic stress. The tissue glucose uptake was high in the weaning period and related to the dominance of *Clostridia* and *Bacteroidia* classes (*Oscillospira*, *Coprococcus*, *RC44*, *Clostridiales*, *Ruminococcaceae*, *Rikenellaceae*, *S24-7* genera) and of metabolic pathways involved in energy-fueling, constitutive growth, and endotoxemia-inflammation-immunity. Maturation affected microbiota composition, reducing systemic inflammation and tissue metabolism, but increasing the susceptibility for liver inflammation, whose microbiota discriminators were *RF39* and *Desulfovibrio* genera. Maternal HFD promoted a proinflammatory microbiota and metabolic pathway profile at both ages, amplifying the weaning reaction (high visceral fat glucose uptake, IL6, liver CT density) and causing adult dysmetabolism (high post-systemic gut and liver glucose uptake in relation to hyperglycemia, with visceral fat being unable to compensate), and monocyte-macrophage dysfunction (low MCP1), increasing proneness for liver injury. We suggest that the visceral network establishes an optimal balance between metabolism and inflammation, and the weaning time window is crucial for its modulation. Our results point to visceral fat as an organ undergoing metabolic stress in early life in the offspring of HFD dams, which may result in a dysfunctional storage capacity in adulthood, and we identify clear imaging biomarkers and bacteria to test the reprogramming of the weaning reaction as a necessary future line of investigation.

## Figures and Tables

**Figure 1 nutrients-13-03438-f001:**
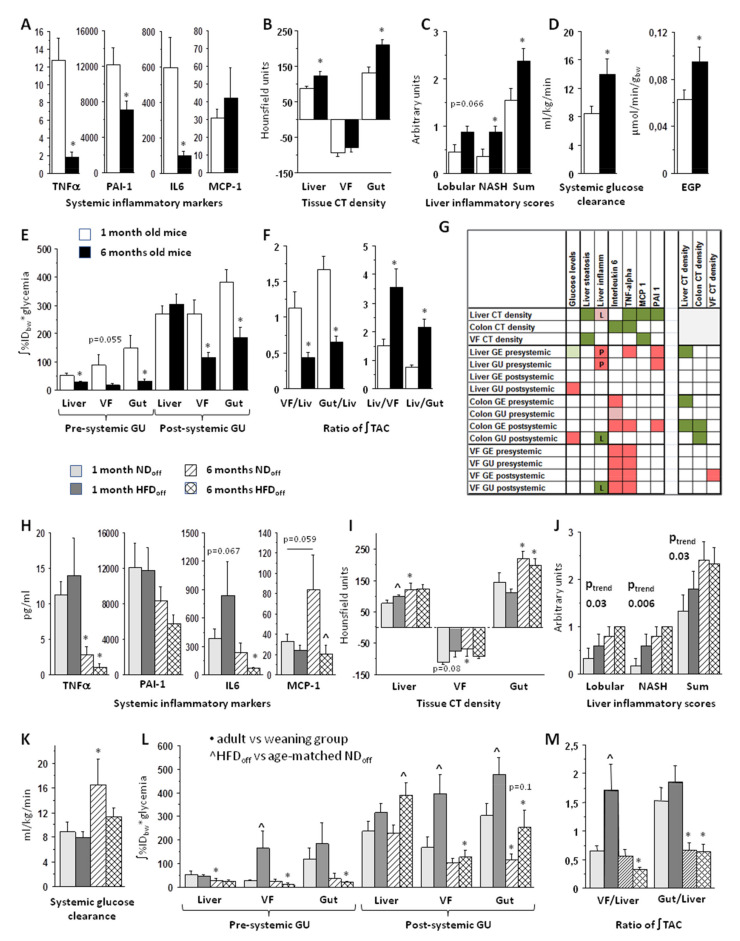
Top panels illustrate the findings according to age-stratified offspring groups (weaning vs. adult mice), showing profound differences in inflammatory markers (**A**), tissue structure by CT (**B**), or liver histology (**C**), systemic glucose metabolism (**D**), glucose uptake rates by PET in liver, visceral fat, gut (**E**), and their ratios (**F**). Panel G documents associations (L = lobular, P = portal inflammation). Corresponding variables are shown in bottom panels (**H**–**M**) in the offspring stratified according to maternal diet*age. GE = glucose extraction, GU = glucose uptake, VF = visceral fat, TAC = time-activity curve, EGP = endogenous glucose production, NASH = non-alcoholic steatohepatitis, CT density refers to attenuation coefficients, i.e., radiodensity in Hounsfield units; * *p* < 0.05 adult vs. weaning group, ^ *p* < 0.05 age-matched HFDoff vs. NDoff; relevant borderline differences are also shown (as text); in (**G**), red (positive), green (negative) associations, dark colors *p* < 0.05, light colors *p* ≤ 0.06.

**Figure 2 nutrients-13-03438-f002:**
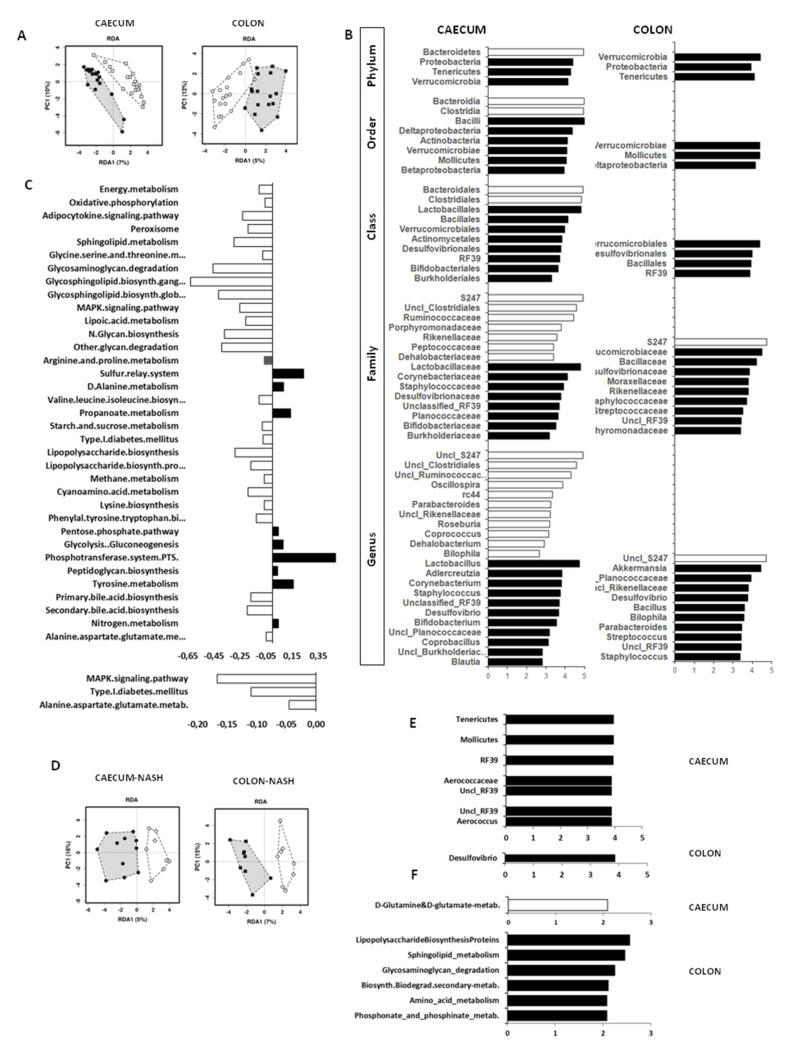
Significant differences in caecum and colon microbiota composition between weaning (white circles) and adult mice (black dots) are shown in RDA analysis (**A**), whereas LDA/LEFSe analyses (**B**) identify unique bacteria taxa biomarkers of weaning mice (white bars) and adult mice (black bars). Significant differences in metabolic pathways from KEGG analysis are expressed as ratios in panel (**C**), in which negative white bars indicate pathways prevailing at weaning, and positive black bars refer to pathways prevailing in adults. Panel (**D**) shows RDA analysis in animals stratified by the presence (black) or absence of NASH (white), and the corresponding significant unique biomarkers are given in panel (**E**) (bacteria taxa) and (**F**) (metabolic pathways).

**Figure 3 nutrients-13-03438-f003:**
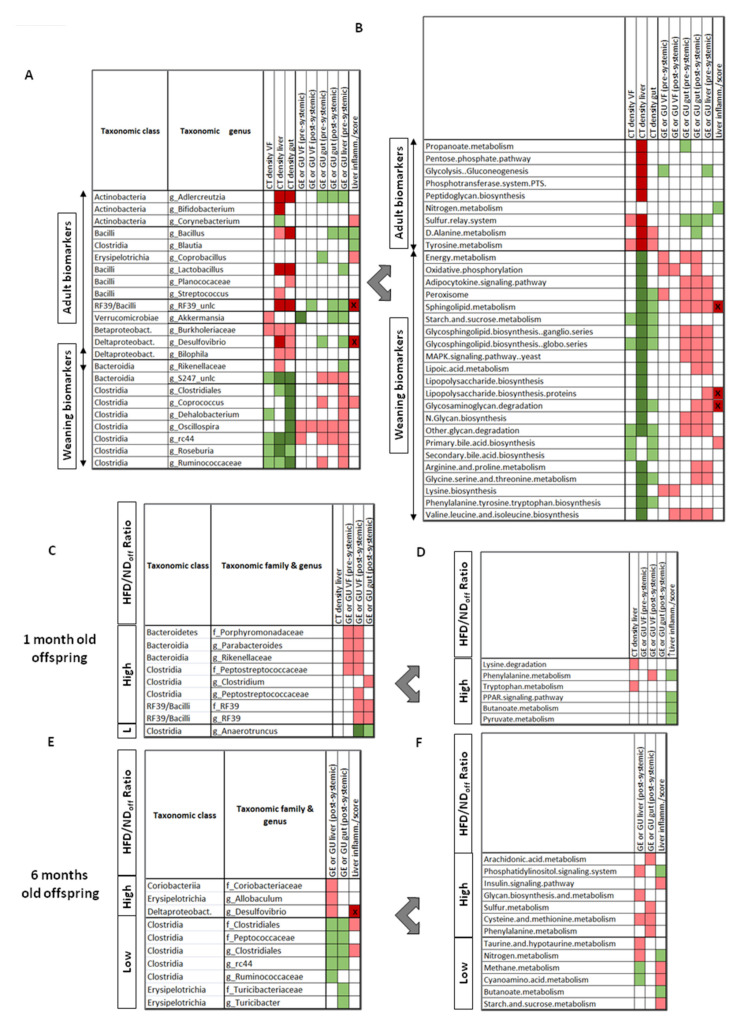
Relationships linking microbiota (left panels) or metabolic pathways (right panels) biomarkers and tissue imaging results (dependent variables), as observed by age (**A**,**B**), or by maternal diet*age (weaning mice (**C**,**D**), adult mice (**E**,**F**)). Significant associations by regression analyses are shown in green (negative) or red colors (positive); darker shades indicate significance after false discovery rate (FDR) adjustment and/or (with the star symbol) in LDA analysis of NASH (from Figure 2). GE = glucose extraction, GU = glucose uptake, VF = visceral fat, CT density refers to attenuation coefficients, i.e., radiodensity in Hounsfield units. Regression coefficients are given in Appendix A to reflect the strength of associations, mostly falling between *r* = 0.4–0.7 for panels A-B, and *r* = 0.5–0.9 for panels (**C**–**E**).

**Figure 4 nutrients-13-03438-f004:**
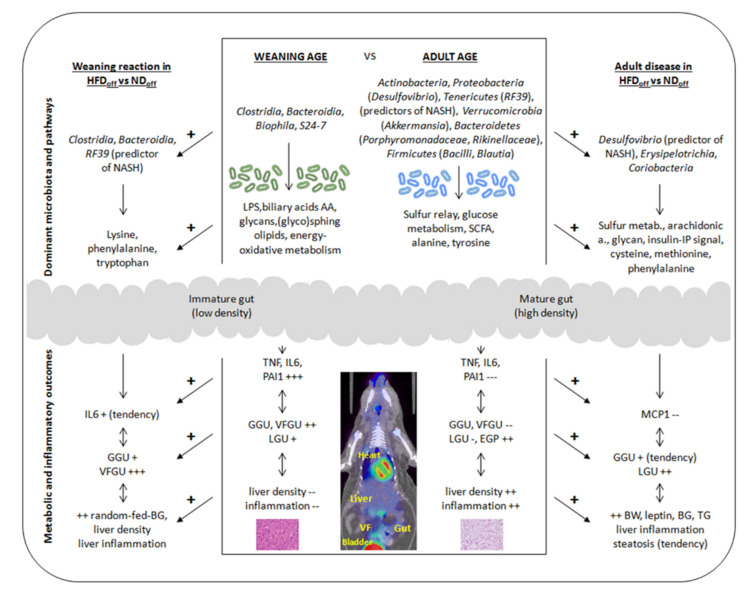
Exemplified diagram summarizing the findings of the study, illustrating changes in microbiota, and systemic and tissue outcomes according to the hypothesized sequence of events, from weaning to adulthood (mid-panel), or accounting for maternal diet at weaning (left panel) or adult age (right panel). GGU, VFGU, LGU = glucose uptake in gut, visceral fat, liver, respectively, BG = blood glucose, and TG = triglyceride levels, EGP = endogenous glucose production, NASH = non-alcoholic steatohepatitis, LPS = lipopolysaccharide, AA = amino acids, SCFA = short-chain fatty acids.

## Data Availability

The data presented in this study are available on request from the corresponding author, as they have not yet been uploaded in a public database.

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
