# Peer review of "Maturation of the Visceral (Gut-Adipose-Liver) Network in Response to the Weaning Reaction versus Adult Age and Impact of Maternal High-Fat Diet"

_nutrients, 2021, doi:10.3390/nu13103438_

Round 1

Reviewer 1 Report

The present manuscript is an extensive work of great interest since its results are a proof of concept of the influence of the mother's diet on the health of the future child. 

Introduction: it should be discussed in detail what the weaning reaction unifies the 18FDG  moenclature, should the 18 go as a super index? See line 108 and 91

V3-V4 region of 16 S rDNA or V3-V4 region of 16S rRNA?

Define CT, other abbreviations in the text such as NASH are not defined.

Figures would look better in a larger size.

Define abbreviations in figures.

A comprehensive review of the English style would be advisable.

Reviewer 2 Report

Review of “Maturation of the visceral (gut-adipose-liver) network in response to the weaning reaction versus adult age and impact of 3 maternal high-fat diet”

This study investigated the effect of maternal diet on the offspring. This study is potentially interesting. However, several problems to be solved.

  1. Many abbreviations, which are not famous, were used, so it was difficult to understand the manuscript. Ex. Abbreviations of VF, GE and GU were difficult to follow. Furthermore, this reviewer cannot understand LV in supplemental Fig 1 and CCA in line 197 mean? To spell out FDR. Line 102 and 103, this reviewer cannot understand NDoff/HFDoff mean? Figure 3, What HFD/ND off ratio mean?
  2. Gut microbiota analysis. The authors mainly compared the difference of gut microbiota between weaning and adult mice in Figure 2. How about the difference of gut microbiota between HFD mother and ND mother?
  3. Figure 1. TNFα, IL6 and PAI-1 were serum or tissue? In addition, what condition these markers measured?
